# Cardiovascular Disease Hospitalizations in Louisiana Parishes’ Elderly before, during and after Hurricane Katrina

**DOI:** 10.3390/ijerph16010074

**Published:** 2018-12-28

**Authors:** Ninon A. Becquart, Elena N. Naumova, Gitanjali Singh, Kenneth K. H. Chui

**Affiliations:** 1Sackler School of Graduate Biomedical Sciences, Tufts University, Boston, MA 02111, USA; Ninon.Becquart@tufts.edu; 2Friedman School of Nutrition Science and Policy, Tufts University, Boston, MA 02111, USA; gitanjali.singh@tufts.edu (G.S.); kenneth.chui@tufts.edu (K.K.H.C.); 3School of Medicine, Tufts University, Boston, MA 02111, USA

**Keywords:** equity, elderly, hurricane Katrina, cardiovascular diseases, segmented regression

## Abstract

The research on how health and health care disparities impact response to and recovery from a disaster, especially among diverse and underserved populations is in great need for a thorough evaluation. The time series analysis utilizing most complete national databases of medical records is an indispensable tool in assessing the destruction and health toll brought about by natural disasters. In this study, we demonstrated such an application by evaluating the impact of Hurricane Katrina in 2005 on cardiovascular disease (CVD), a primary cause of mortality among older adults that can be aggravated by natural disasters. We compared CVD hospitalizations before, during and after Katrina between white and black residents of three most populated parishes in Louisiana: Orleans and Jefferson, which were severely affected by the landfall and subsequent floods, and East Baton Rouge, which hosted many of the evacuees. We abstracted 383,552 CVD hospitalization records for Louisiana’s patients aged 65+ in 2005–2006 from the database maintained by the Center of Medicare & Medicaid Services. Daily time series of CVD-related hospitalization rates at each study parish were compiled, and the changes were characterized using segmented regression. In Orleans Parish, directly affected by the hurricane, hospitalization rates peaked on the 6th day after landfall with an increase (mean *±* SD) from 7.25 ± 2.4 to 18.5 ± 17.3 cases/day per 10,000 adults aged 65+ (*p* < 0.001) and returned to pre-landfall level after ~2 months. Disparities in CVD rates between black and white older adults were exacerbated during and following landfall. In Orleans Parish, a week after landfall, the CVD rates increased to 26.3 ± 23.7 and 16.6 ± 11.7 cases/day per 10,000 people (*p* < 0.001) for black and white patients, respectively. The abrupt increase in CVDs is likely due to psychosocial and post-traumatic stress caused by the disaster and inadequate response. Inequities in resource allocation and access have to be addressed in disaster preparation and mitigation.

## 1. Introduction

The United Nation World Conference on Disaster Risk Reduction held in Sendai, Japan on 14–18 March 2015 enforced the Hyogo Framework for Action, approved by the 168 member-states in 2005, and presented the Sendai Declaration aiming to further improve the multipronged response to disasters [1]. The Declaration paid special attention to building community resilience, engaging and empowering vulnerable populations, including children, elderly, and individuals with special needs. The *Bloomberg Business* Magazine recently reported the cost of damages from Harvey, Irma, and Maria hurricanes at about $202.6 billion [2], almost 100 times of $3.2 billion funds that the Federal Emergency Management Agency (FEMA) has obligated to public assistance after Hurricane Katrina to repair and rebuild public infrastructure, such as bridges, roads, schools, hospitals and sewer treatment facilities [3,4,5,6].

Hurricane Katrina was a harrowing experience for those impacted on 29 August 2005, when it hit the Gulf Coast. Homes were destroyed, two million people were displaced, and survivors experienced health effects that could be long-lasting [7]. Orleans, Jefferson, and St. Bernard parishes were the most affected, containing 95% of Katrina’s victims, and approximately 80% of New Orleans was flooded [8]. A 2011 technical report by the National Oceanic and Atmospheric Administration (NOAA) stated that Hurricane Katrina was the costliest hurricane of all time, and the third most deadly and intense since 1900 [9]. It set records for the greatest count of tropical storms and hurricanes and tied the record for the most major hurricanes in a 2-year period [9].

The storm disproportionately affected older adults, black Americans and low-income communities. There were an estimated 971 deaths in Louisiana as a result of Hurricane Katrina, and the earliest and most numerous deaths were seen in the elderly [8,10]. Almost half of all the deceased were older than 75 years old, and an additional 26% were of the ages 60 to 75 [11]. Besides deaths, the vulnerable elderly people were substantially overrepresented among the evacuees—a population potentially predisposed to a high level of psychological distress exacerbated by severe disaster exposure, lack of economic and social resources, and inadequacy in government response [9]. In Orleans Parish, the mortality rate for black residents aged 18 or order was 1.7 to 4 times greater than that of the white [8]. In New Orleans, damaged areas had higher proportions of black residents and poor households than undamaged areas (45.8% versus 26.4%, and 20.9% versus 15.3%, respectively) [7]. Possible reasons for such post-disaster racial health disparities included insufficient supply of medication and limited access to need resources. Analyses revealed that black respondents had a limited access to a 3-day supply of their medications and this difference could be due to prescription drug coverage in health insurance plans, as well as access and proximity to pharmacy [12]. 

The most recent hurricane seasons repeatedly demonstrated that the needs of the elderly population in the United States during and following a disaster were often neglected [13]. The study of nationally representative panel survey of older Americans indicated that older adults in lower income status had lower preparedness level than those in higher income [14]. Most disaster response models focus on acute injury treatment, environmental risk management, and infectious disease prevention [15]. However, interviews and focus groups with both providers and survivors’ contrast with the need to also include continued care for survivors with chronic conditions such as cardiovascular disease [15]. Among the Katrina survivors, 41–74% had one or more chronic conditions [15]. These survivors could not obtain medications due to availability or affordability, were unable to access medical records onsite or at a distance, and had limited healthcare resulting from the most significant efflux of the healthcare workforce known in U.S. history [7,15]. It is also possible that CVD disproportionally affected black survivors: in general, compared to white patients, patients of color experience higher CVD rates and risk factors, encounter more barriers to diagnosis, are given lower quality healthcare, and endure worse health outcomes [16]. In addition, black patients experience lower cardiac intervention rates, lower admission rates, and shorter cardiac care unit stays compared to white patients [16]. 

The complex nature a disaster typically triggers a series of cascading responses peaking at different times and forming a unique signature of each event. Such signatures have been observed in many studies analyzing the health effects of natural disasters. A study analyzing the effect of a 2011 tsunami on acute decompensated heart failure (ADHF) reported a peak in the weekly number of ADHF cases 3 to 4 weeks following the tsunami, after which number of cases per week returned to pre-disaster level [17]. Acute coronary syndrome, stroke, and cardiopulmonary arrest, pulmonary thromboembolism, and infectious endocarditis cases exhibited both a rapid increase and decline, whereas heart failure cases remained significantly increased for up to 7 weeks after the earthquake [18,19]. Examining the consequences of Hurricane Sandy in 2012, researchers have detected increases in hospitalizations and deaths due to myocardial infarction and stroke during the 2 weeks after landfall, compared to the same 2 weeks from the 5 previous years [20]. An increase of acute myocardial infarctions (AMI) admissions during 3 years after Hurricane Katrina, as compared to the 2-year period before the storm was noted [21]. Another study demonstrated that, during the 6 years following Katrina, the number of admissions for AMI tripled [22]. Researchers inferred that the higher incidences of severe cardiovascular events observed during natural disasters could be attributed to elevated stress [23], greater physical activity, change in blood rheology, diminished attention or ability to address medical needs, and insufficient emergency services [20]. Researchers credit lack of food and medications, which were also a reported problem following Hurricane Katrina, a possible cause [17]. 

The difficulties with collecting information during and after disasters are well recognized; underserved communities typically suffered due to limited resources allocated to data collection. The research on how health and health care disparities impact response to and recovery from a disaster, especially among diverse and underserved populations is in great need for thorough evaluation. The utilization of national databases of medical claim of patients had illustrated a strong potential in assessing environmental impacts on vulnerable populations [24,25,26]. The Center of Medicare & Medicaid Services (CMS) has been maintaining uniformly collected hospitalization records with a coverage rate of ~98% of US adults aged 65 years old and older. The large scale and rapid progression of natural disasters limit opportunities for detailed data collection over time, yet the time series analysis of medical records can help in recreating the short-term and long-term responses at a refined temporal and geographical scale [27,28].

The objective of this study is to compare the daily rates of CVD hospitalizations before, during, and after Hurricane Katrina among older adults of Louisiana using CMS records. We established the critical periods in CVD responses to Katrina and examined hospitalization rates for black and white residents in three communities using segmented regression models adapted to time series analysis. We selected three parishes (equivalent to “counties” in other states): Orleans and Jefferson, two densely populated parishes most affected by the landfall and subsequent floods due to being on the storm track as well as East Baton Rouge—the parish that accommodated many evacuees [29]. We hypothesized that there was a significant difference in the number of cases, rates, and in the demographic profiles for the different periods. We also hypothesized that black elderly adults would be disproportionately impacted by the storm, as seen through higher CVD rates compared to white adults. While studies on the effects of natural disasters, including earthquakes on CVD exist, those focusing on hurricanes are sparse [20]. This is therefore one of the few studies analyzing the effect of hurricanes on CVD specifically. Results from this study can help further our understanding of natural disaster impacts on CVD and help health authorities better prepare for upcoming seasonal events. Our long-term goal is to pave the way for a solid methodology in assessing the impact of a disaster in order to better understand the cardiovascular needs of our elderly population and to ultimately advance disaster planning and long-term recovery efforts. 

## 2. Materials and Methods

### 2.1. CVD Records

Medical claims were obtained from the Centers for Medicare and Medicaid Services (CMS). For the USA, 21,363,136 records of CVD hospitalizations were retrieved from the database in SQL Server 2014 Management Studio for all relevant *International Classification of Diseases, Ninth Revision, Clinical Modification* (ICD-9M CM) codes in any of 9 diagnostic fields for patients aged 65 and older from 1 January 2005 to 31 December 2006. The codes included: 390–398, 401–407, 420–438, 440–449, 451–459. In Louisiana, there were 383,552 hospitalizations, including 119,612 cases with CVD listed as primary diagnoses. The distribution of specific sub-codes is shown in Appendix A
Table A1. Of all the selected records, 17,769 were for Orleans Parish, 21,499 for Jefferson Parish, and 24,699 for East Baton Rouge Parish. Each record contained information on date of admission, sex, age, race, county and zip code of residence. Based on the county code of residence, we compiled counts for each county in the state of Louisiana. (Note, that in Louisiana, a “parish” is equivalent to a “county,” which is a more generic term, so both terms can be used interchangeably.) We intentionally removed the first 6 days and the last 12 days to reduce the error associated with incomplete reporting at the beginning and at the end of the study period. Therefore, we reduced the study period to 710 days from 7 January 2005 to 17 December 2006. Based on the date of admission and patient’s race, we compiled time series of daily counts for the study period for three selected counties or parishes (Orleans: 17,394 cases, Jefferson: 21,057, and East Baton Rouge: 24,147 cases) for black, white and all adults aged 65 or above (see Appendix A
Table A2 for an example of the first 10 rows; full data set is available on request).

### 2.2. Census Data and Hospitalization Rates

Census data for the population 65 years and older on the County level was obtained from Census.gov for years 2005 and 2006. Because these are not Census years, American Community Survey (ACS) estimates and Intercensal estimates were used to supplement the Census data [30]. Hospitalization rates were calculated for spatial mapping and temporal analysis. For spatial mapping county-specific hospitalization rates were calculated by dividing counts of CVD cases per day for all period, pre-Katrina and post-Katrina periods by the parish population (65+) with the multiplier of 100,000. For temporal analysis, for each of the three parishes the daily hospitalization rates were calculated by dividing counts of CVD cases per day by the interpolated estimates of parish population with the multiplier of 10,000. The 2005 population counts were used for the period from 7 January 2005 to 29 August 2005, the day that Hurricane Katrina struck the Gulf coast. Hurricane Katrina caused a mass exodus of people from Orleans Parish and the surrounding area. Following Hurricane Katrina, it was more reasonable to use the 2006 population counts to reflect the loss or gain in population in a parish. 

### 2.3. Mapping

We mapped the state of Louisiana populations by county, population density, daily hospitalization cases, and average daily hospitalization rates using ArcGIS 10.3.1 (Esri, Redlands, CA, USA). Because ACS estimates were not available for every parish in Louisiana, maps were produced with intercensal estimates for the whole study period, as well as pre- and post-Katrina’s landfall periods (see Appendix A
Table A3). Storm track was mapped using data from the Mississippi Automated Resource Information System (MARIS).

### 2.4. Statistical Analysis

*Identifying critical periods using a loess smoother:* A part of preliminary analysis, we superimposed daily time series of rates with a loess smoother [31] to assess the overall trend and weekly pattern: (1)$$y_{t}=\beta_{0}+\beta_{1}lo\left(t,\alpha \right),$$
where $y_{t}$ is CVD hospitalization rate at day t, $\beta_{0}$ represents the intercept, and $\beta_{1}$ represents the slope of the smoothed regression line. The study days were numbered from 1 to 710 (the total number of days being analyzed in the time series), starting from 7 January 2005 and ending on 17 December 2006. The span parameter α of the loess smoother *lo(.)* was selected to be equal to 7/710 to reflect the weekly pattern that has also been previously reported for admissions due to AMI, a CVD subtype [22]. From the analysis of the Orleans Parish residents we identified six time period of interest to characterize the periods before evacuation, before the landfall, immediately after landfall, and recovery periods, which are also supported by reports about Katrina [32,33]. 

*Segmented regression model:* The study time period of 710 days divided into six intervals (T1, T2, T3, T4, T5, T6) were marked with knots, or endpoints: *a*, *b*, *c*, *d*, *e* at days 225, 238, 247, 275, and 303, respectively. Using the selected periods, we develop a segmented linear regression model [34], as follows:(2)$$E(y_{t})=\beta_{0}+\beta_{1}t+\beta_{2}{\left(t-a\right)}_{+}+\beta_{3}{\left(t-b\right)}_{+}+\beta_{4}{\left(t-c\right)}_{+}+\beta_{5}{\left(t-d\right)}_{+}+\beta_{6}{\left(t-e\right)}_{+},$$
where
(3)$${\left(u\right)}_{+}=\{\begin{array}{c} u,\;u>0,\\ 0,\;u\leq 0. \end{array}$$

Thus,
$$\begin{array}{ccc} E(y_{t}) & =\beta_{0}+\beta_{1}t, & t\leq a\\ & =\beta_{0}+\beta_{1}t+\beta_{2}\left(t-a\right) & a<t\leq b\\ & =\beta_{0}+\beta_{1}t+\beta_{2}\left(t-a\right)+\beta_{3}\left(t-b\right) & b<t\leq c\\ & =\beta_{0}+\beta_{1}t+\beta_{2}\left(t-a\right)+\beta_{3}\left(t-b\right)+\beta_{4}\left(t-c\right) & c<t\leq d\\ & =\beta_{0}+\beta_{1}t+\beta_{2}\left(t-a\right)+\beta_{3}\left(t-b\right)+\beta_{4}\left(t-c\right)+\beta_{5}\left(t-d\right) & d<t\leq e\\ & =\beta_{0}+\beta_{1}t+\beta_{2}\left(t-a\right)+\beta_{3}\left(t-b\right)+\beta_{4}\left(t-c\right)+\beta_{5}\left(t-d\right)+\beta_{6}\left(t-e\right) & e<t. \end{array}$$

This long expression can be summarized as follows:(4)$$E\left(y_{t}\right)=\beta_{0}+\beta_{1}t+\sum_{t=1}^{t}\beta_{t+1}{\left(t-{\left(u\right)}_{+}\right)}_{+},$$
where ${\left(u\right)}_{+}$ are the locations of the break points, $y_{t}$ is CVD hospitalization rate at day t. The regression coefficients $\beta_{1}$ and $\beta_{2+}$ are the slopes for the selected time intervals.

*Sensitivity analysis with a broken-stick model:* As part of sensitivity analysis we estimated the intercept (or the value at the critical endpoints) and slope values using a broken-stick model for each pair of adjacent periods [35], as follow:(5)$$E\left(y_{t}\right)=\beta_{0}+\beta_{1}t_{1}+\beta_{2}t_{2},$$
where $y_{t}$ is daily CVD hospitalization rate, $t_{1}$ and $t_{2}$ represent the time for adjacent intervals, $\beta_{0}$ is the intercept, and $\beta_{1}$ and $\beta_{2}$ are the slopes for the adjacent time intervals. The results of the models along with the quality of fit are presented in Appendix A
Table A4. The segmented model and the broken-stick model were run for total, black and white patients for each parish. All data processing and model building was done using *R*.

## 3. Results

For the 710-day study period, there were 383,552 hospitalizations due to CVD in Louisiana, including 119,612 cases with CVD listed as primary cause of hospitalization (Appendix A
Table A1). The 420–429 ICD codes, referring to acute pericarditis, endocarditis, myocarditis, cardiomyopathy, conduction disorders, cardiac dysrhythmias, heart failure, and other complications of heart disease represented the majority (40%) of CVD types for all subpopulations. For black patients, hospitalizations due to hypertensive heart diseases (401–405 ICD codes) were twice higher than for white patients (9.8% vs. 4.7%), in contrast with ischemic heart disease.

ACS-derived demographic data for each parish, or county of the state of Louisiana were mapped (Figure 1; data are provided in Appendix A
Table A3). Figure 1 shows geographic distribution of population aged 65 and older and the average number of daily CVD cases per county (Panel A) and a close-up for the Orleans, Jefferson, and East Baton Rouge Parishes (Panel B). These three parishes had high population density before and after Katrina’s landfall (Periods D and F). Orleans Parish and Jefferson Parish were closest to the storm track and had the highest number of people affected by the hurricane.

### 3.1. Identification of Critical Periods

We identified six time periods of interest using loess-smoothing technique as shown in Table 1 and Figure 2 and examined the rates for each selected parish accordingly. The daily time series of daily CVD hospitalization rates per 10,000 elderly adults (shown as points) with the superimposed loess-smoothed curve (shown as line) are depicted in Figure 2 for Orleans Parish (Panel A), Jefferson Parish (Panel B), and East Baton Rouge Parish (Panel C), respectively. The landfall occurred on 29 August 2005 was indicated by a border between Periods T2 and T3, right after day 240. In Figure 2, the different hues and capital letters (T1–T6) indicate six distinct time periods according to Table 1.

In Figure 1 we mapped CVD counts and rates corresponding to the time periods before (Period T1, Panels C and D) and soon after the storm (Periods T3 and T4, Panels E and F). Prior to the storm the number of older residents in the East Baton Rouge Parish was comparable with the Orleans Parish. After the storm it received evacuees [29], including over five thousand older adults. Although St Bernard Parish also had a high rate of CVD, however due to a low population density the number of hospitalization records were insufficient for the further analysis.

As shown in Table 2, Orleans Parish had a notably high average daily CVD hospitalization rate of 17.0 ± 10.23 cases/day per 10,000 people aged 65+ after the storm (Periods T3 and T4), while the number of residents declined by almost 17,000 people. In East Baton Rouge the total number of older adults increased by almost 5000 people, although with some decline in CVD rates.

### 3.2. Segmented Regression Analysis

The time series of daily rates with the superimposed fit from the segmented regression for each of six periods and for all three parishes are shown in Figure 3 (Panels A, B and C). As shown in Table 3, In Orleans Parish, the average CVD hospitalization rates was 7.25 cases/day per 10,000 persons and was decreasing slowly for over 6 months before Katrina (Period T1). During the evacuation week (Period T2), it dropped to 3.83 cases/day (Tukey-test, *p* = 0.002, as compared to T1). In the week following Katrina (Period T3), the average daily rate increased drastically to 21.18 (*p* < 0.0001) by 3.12 additional cases per day. 

The rate reached its peak of 45.36 cases/day on Sunday, 4 September, 6 days after Hurricane Katrina’s landfall. In the following month (Period T4) the daily rate remained significantly high 15.52 cases/day (*p* < 0.0001) compared to pre-storm, pre-evacuation Period T1, but was decreasing by 3.56 cases per day. Two months after the landfall (Period T5) the average rate returned to a pre-storm, pre-evacuation level and then started to decline in Period T6 in manner similar to Period T1.

As shown in Table 3, in Jefferson Parish the hospitalization rate was increasing from an average rate of 5.01 cases/day during evacuation Period T2, one week before landfall, to 8.18 cases/day in a week after landfall (*p* < 0.05, compared to T1). The highest rate 12.7 cases/day was observed on Saturday 3 September, 5 days after Hurricane Katrina’s landfall. The month following the peak, rates remained significantly high, at 7.26 cases/day (*p* = 0.001). Two months after, average rate returned to 5.36 cases/day, similar to the pre-evacuation pre-storm Period T1 (*p* = 0.9). 

In contrast with Orleans, in East Baton Rouge Parish while the average hospitalization rates remained stable over all study periods except for a week after the landfall Period T3, when the rates were rapidly declining by 0.47 cases per day (*p* < 0.001). In the month following landfall, the average rate of 6.55 cases/day was a significantly lower as compared to 8.69 cases/day per 10,000 persons in Period T1 (*p* = 0.003). During the evacuation and after the landfall (Periods T2–T6), rates in East Baton Rouge Parish appeared to change in the opposite direction of Orleans Parish and Jefferson Parish.

### 3.3. Racial Differences in CVD Rates

When average hospitalization rates are calculated separately for black and white older adults for two main periods: pre-Katrina (Period T1) and during periods T3 and T4: starting the day after landfall and ending 28 days after the peak response (see Table 2), several differences are revealed. The proportion of black residents in Orleans Parish was higher than in Jefferson and East Baton Rouge Parish before and after the storm, yet the reduction in population in Orleans Parish was almost twice as large among black residents than white residents. In all parishes the racial differences in CVD rates were notable either before or after the storm (Table 2 and Table 3, Figure 4). In Orleans Parish, the highest CVD rates, which stand out as outliers about a week after landfall, are all for black adults. They experienced a significant increase in rates soon after landfall of 26.3 and 17.5 cases/day in Periods T3 and T4, respectively (*p* < 0.001, Table 3). The direct comparison of average daily rates between black and white older adults post landfall (19.8 ± 13.7 vs. 14.5 ± 7.8, *p* < 0.05, Table 2) shows significantly higher rates black patients. In Jefferson and East Baton Rouge Parish, the CVD rates were significantly higher in black than in white older adults before the storm.

In the most affected Orleans Parish, white adults also experienced significant increases immediately after the storm reaching 16.6 and 13.8 cases/day in Periods T3 and T4, respectively (*p* < 0.02, Table 3) but of a lesser extent. The week following landfall, each day brought new hospitalizations: on average 3.96 ± 0.25 for blacks and 2.32 ± 0.20 for whites. During the post-Katrina period of stabilization and recovery in all parishes the average daily rates in both black and white adults were consistently lower than before the storm.

## 4. Discussion

Our study demonstrates significant changes in CVD hospitalization rates in Orleans Parish, both during and following Hurricane Katrina’s landfall. On Friday, 26 August, two days before the landfall, the governor of Louisiana, Kathleen Blanco, declared a state of emergency [3]. On Saturday, 27 August, Orleans Parish, Jefferson Parish and St. Bernard Parish ordered voluntary evacuations, recommending that all residents evacuate, particularly those living in lower areas. Mandatory evacuation was issued the day before landfall, on Sunday, 28 August [3]. On Wednesday 31 August, two days after landfall, New Orleans mayor Ray Nagin estimates that 50,000 to 100,000 people remain in the city, which officials say is 80% flooded [31]. Our analysis demonstrated that during evacuation period the daily rates of CVD had significantly decreased in Orleans Parish. The rates then spiked immediately after Katrina’s landfall in all older adults, especially in black residents. The rates appear to stabilize about two months after the landfall, coinciding with the clearing of floodwaters from New Orleans and the end of the emergency period, which lasted from the disaster event to the “dewatering” of New Orleans.

The surge in CVD observed in Louisiana after Hurricane Katrina could be due to many apparent reasons. Psychosocial stressors, lack of medication and proper support system are likely to play a key role. After Katrina, multiple reports have indicated high rates of psychosocial stress and post-traumatic stress, as well as an almost threefold increase in suicide rates [22]. Survivors had to cope with stressors such as searching for food and shelter, relocating, crowding, being in financial hardship, and dealing with insurance companies and social services [11]. These stressors not only reminded survivors of their trauma, but also introduced negative impacts to their capacity to cope [11]. This is a specific health concern given that psychological stress has emerged as a major risk factor for cardiovascular disease [36]. Stressors such as hurricanes and other natural disasters can cause chronic and acute mental stress, which can then trigger cardiovascular events [37]. After all, traditional cardiac risk factors only account for half of the causes of CVD, with most of the remaining risk explained by psychosocial factors [37].

There was also an estimated doubling of mental illness incidences following the storm: survivors who relocated to Oklahoma exhibited distress, disability, and post-traumatic stress disorders (PTSD) symptoms [38]. Survivors had an increased and abnormal baseline heart rate compared with that of controls, especially for those with depressions. Some survivors had decreased heart rate reactivity and heart rate variability, which are independent risk factors for cardiovascular disease [38]. In addition, an earlier study by the same group found immune system changes in a group of young, healthy Katrina survivors, stating that in the presence of PTSD, they had increased IL-6 levels, with unknown cardiovascular health complications [39]. In fact, heart conditions were reported to be one of the top three major causes of death among Louisiana victims [8].

Acute CVD events are known to be triggered by sudden emotional or physical stressors, such as unanticipated natural disasters, which are known for causing some of the strongest acute and subacute types of psychological stress [40]. It is likely to be the case for our study, when the majority of hospitalizations were due to acute conditions such as pericarditis, endocarditis, myocarditis, cardiomyopathy, cardiac dysrhythmias, and heart failure. The impact of psychosocial stress on cardiovascular health can be understood through the concept of allostasis—the ability to maintain stability through change, allowing one to respond to environmental demands, such as shifts in our physical state or dealing with stressful situations [40,41]. Both the sympathetic nervous system and hypothalamus-pituitary-adrenal (HPA) axis are involved in allostatic responses [40]. The adrenal medulla and sympathetic nerves release catecholamines when the sympathetic nervous system or HPA axis are activated, and the hypothalamus releases corticotrophin-releasing hormones that mediate the pituitary’s release of corticotrophin, and thus, the adrenal cortex’s release of cortisol [40]. Allostatic load then results from the allostatic mediator hormones changing cellular physiological and pathophysiological responses. Stress can trigger allostatic load, which can eventually, in turn, trigger cardiovascular disease [40]. It is very possible that Hurricane Katrina, which was both a sudden catastrophic event, and thus a source of both immediate and chronic stress, could have triggered allostatic load, which then led to the sharp increase in CVD rates observed after landfall, both immediately and even a month later.

It is worth noting that mortality rates for black adults was 1.7 to 4 times higher compared to rates for white adults in Orleans Parish [8]. Higher mortality rates were also seen for black adults 75 years and older compared to white adults of the same age group [8]. Low-income households, minority households, and households with elderly or disabled people are also less likely to evacuate than other households [42]. The already-existing gap in CVD rates between black and white older adults, has predisposed blacks to be more greatly impacted in the aftermath of the storm. It may appear that, perhaps the black adults were experiencing CVD sooner and at higher rates, that on average not as many survived, especially those prone to CVD; thus, rates of CVD deaths for the black population fell in the months after due to the loss of these individuals. Another possibility could be that access to healthcare and other resources differed significantly for these two racial groups following the storm, so black individuals were still experiencing CVD events, but were faced with limited hospital access [43]. Studies have indicated that black individuals have a higher likelihood of experiencing PTSD following a hurricane, compared to white individuals, and reported a greater impact of loss of services compared to white adults [43]. Yet another explanation could be that black residents lived in neighborhoods that were affected by the storm at a greater magnitude, and thus left and returned to Orleans at lower rates than the white population, especially soon after the storm. Damaged areas were also more likely to be comprised on households with low socioeconomic level. Both race and class are intertwined in this case as a part of the residential segregation reported in Orleans Parish, which results in polarized social capital for those groups as well as access to different institutions regarding emergency preparedness [7,44]. Stakeholders should take this into account and ensure access to medical care and emergency preparedness for all individuals, regardless of neighborhood, socioeconomic status, and race.

In East Baton Rouge Parish, CVD rates were higher on average for black adults, confirming what has been reported in literature previously [16,45,46]. East Baton Rouge Parish received evacuees [29], no significant changes in CVD hospitalization rates were noted, and the pattern was somewhat reversed to the observed in Orleans. This lack of detected change could exist for several reasons, including lack of tracking patient mobility and location of receiving care. First, in comparing rates across parishes, racial populations and time periods we rely on the population estimates that might be not accurate. Typically, the Intercensal estimates are updated yearly, and thus do not capture well fluctuations in population due to disaster related migrations, displacement, and losses. Furthermore, by focusing on the elderly population, which is already a subset of the evacuee population, we are not able to detect the impact on population younger than 65 years old, who yet could overwhelm health care facilities. There were not only people evacuating out of New Orleans, but also evacuees who returned eventually, as well as an influx of new people into the region due to construction jobs following the storm. From East Baton Rouge Parish, some evacuees also returned to Orleans Parish, especially due to its close proximity compared to other cities that accommodated many evacuees, such as Atlanta or Dallas. 

We merged population data from American Community Survey estimates and Intercensal estimates to supplement the census data and account for a mass exodus from affected area; however, it would be more assuring to have reliable public resources to properly account for changes in population following disaster. Such data repositories should serve many purposes, including disaster mitigation and relieve and assessing the quality and efficiency in controlling the impact of a disaster. The study conducted by the Centers for Disease Control and Prevention soon after Hurricane Katrina revealed that chronic disease and related conditions, including cardiovascular and cerebrovascular diseases accounted for a significant proportion of visits to emergency treatment facilities in and around New Orleans, particularly among people ages 60 years or older [47]. This study led us to think about importance of better harmonization of all available data, including hospitalization claims and surveillance records [48].

This study used patient’s county of residence for billing purposed and is unlikely to track well where patients received their care. This phenomenon was observed in studying hospitalizations due to seasonal pneumonia and influenza using CMS claims [49]. We had utilized the information contained in the CMS records represents that of the beneficiary residency, and not the provider. Survivors from Orleans Parish or Jefferson Parish may have received care in East Baton Rouge Parish, but this type of information is not readily available. Due to lack of knowledge how reliably a patient’s residential address, recorded by the CMS database, reflect the actual patient location, we roughly assessed hospital utilization. We estimated that before the Katrina’s landfall 97% of Orleans Parish patients were treated in Louisiana; during and two months after the landfall, 45–56% received health in hospitals outside of Louisiana; and afterwards 78% of patients with Orleans Parish county code of residence were treated in Louisiana. The access and utilization of healthcare facilities could have changed during disasters and we were not able to account for such changes. This crucial information is often scattered among various agencies and not readily available. 

This report is one of the few papers discussing the impact of hurricanes on CVD hospitalization rates, as well as the disparities in rates between white and black elderly adults. We offered a novel application of segmented regression model to explore in detail a complex dynamic process with rapid rise and fall in CVD hospitalizations. Further research is recommended to identify patterns in CVD hospitalizations by subtype, as well as any differences in peak intensity between subtypes. We used Medicare records, which provide a broad coverage for the majority of American adults 65 years and older. While this data set is uniquely useful, it could not be used to provide timely reports during and after emergency, as it requires at least 6 months to request and assemble the records. It is expensive to operate and maintain even for experienced researchers. Yet, hurricanes are striking seasonally on a regular basis, so a solution for timely and reliable reporting is critical to develop proper preparedness strategies for hurricanes, whose impacts are consistently intensifying in the region. The presented findings provide the basis for planning resource allocation, among other disaster needs.

There are many questions we were still unable to address: What were the mechanisms underlying the disparities? Why did we see the disparities? What other forms of disparity occurred beyond race? After the event, did the disparity go away? Did the recovery for each group occur with the same speed or did difference occur in recovery by racial groups? Studies suggest that preparedness of black elders was not significantly different from that of whites; however, older black adults in lower income status were significantly less prepared for disaster than other groups [14]. This study only examined records 16 months post-Katrina, however the recovery from economic, physical and psychological damages may last for many years. The impact caused by hurricane damage, loss, or stress can be alleviated by housing, finance, and stress management. Studies of Kobe earthquake survivors show that “through event evaluation, social ties and community rebuilding efforts directly or indirectly facilitated the reframing of earthquake experiences into positive narratives” [50]. The reasons for the steady decline in CVD rates post-Katrina remain unclear; it might be due to older adults who were frail moved to other counties or states. We recommend that policies and programs for post-disaster evaluation should also consider the factors of family- and community-level enrichment, empowerment, and commemoration, allowing for the transcending and effective communication of the experience, ultimately leading to lasting improvements. Recovery periods could create opportunities for rebuilding infrastructure and reframing the social relationships, with proper policies and procedures these periods can serve as an optimal juncture to reduce racial, economic and environmental inequalities.

## 5. Conclusions

This study analyzed changes in elderly CVD hospitalization rates occurring during various time periods before, during, and after Hurricane Katrina in Orleans, East Baton Rouge, and Jefferson parishes, all of which are located near the storm’s track. CVD hospitalization rates rose immediately after landfall in both Orleans and Jefferson parishes. This increase in CVD hospitalization rates was prolonged, lasting more than a month after landfall. An increase in CVD rates was higher among elderly black populations compared to elderly white populations in the weeks after landfall, indicating differences in level of impact in these two communities. The 2017–2018 hurricane seasons were particularly devastating, as storms like Harvey, Irma, Maria, and Jose ravaged major metropolitan hubs in Houston, Puerto Rico, and Dominica and prolong disasters of Florence and Michael in Carolinas and Florida. Considering the staggering economic and health impacts of such disasters, lessons from the past hurricanes, including Katrina experience should inform future health management decisions.

## Figures and Tables

**Figure 1 ijerph-16-00074-f001:**
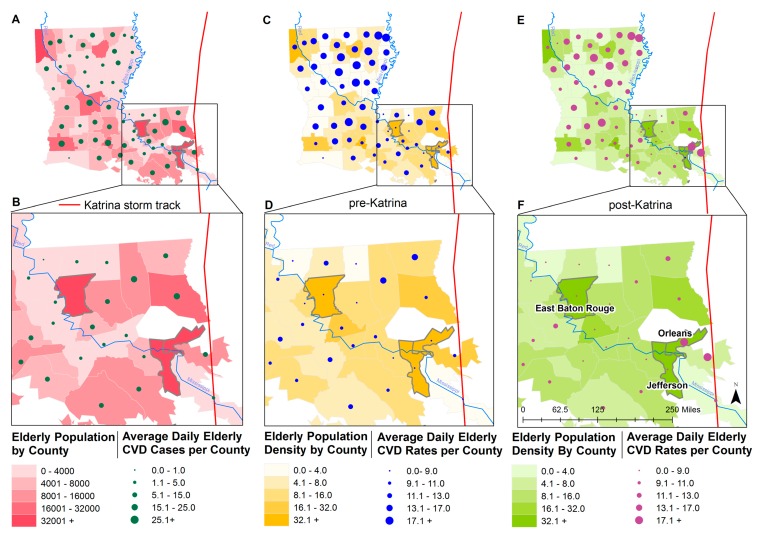
Geographic distribution of older population and average daily cardiovascular disease (CVD) cases per county shown for the state of Louisiana (**Panel A**) and a close-up showing Orleans, Jefferson, and East Baton Rouge Parishes (**Panel B**) for the whole study period (Periods T1–T6). Geographic distribution of population density (population/mi^2^) and average daily CVD rates before the storm (Period T1) are shown for Louisiana (**Panel C**) and close-ups of three study parishes (**Panel D**), and after the storm (Periods T3 and T4), for Louisiana (**Panel E**) and a close-up of three study parishes (**Panel F**).

**Figure 2 ijerph-16-00074-f002:**
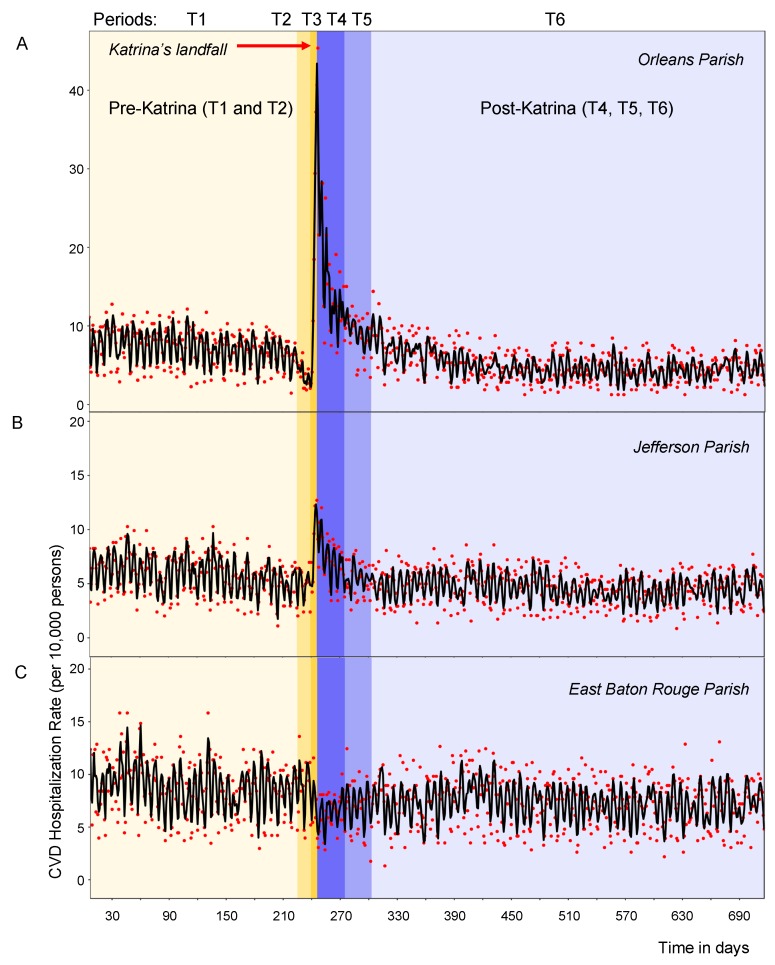
Time series of daily CVD hospitalization rates per 10,000 elderly adults with superimposed smoothed values for 7 January 2005–17 December 2006 in Orleans Parish (**Panel A**), Jefferson Parish (**Panel B**), and East Baton Rouge Parish (**Panel C**). Actual daily hospitalization rates are displayed as points; rates predicted by a loess-smoother are displayed as lines. The different hues and capital letters indicate six distinct time periods with landfall occurred on 29 August 2005 right after day 240, at the border between T2 and T3 periods.

**Figure 3 ijerph-16-00074-f003:**
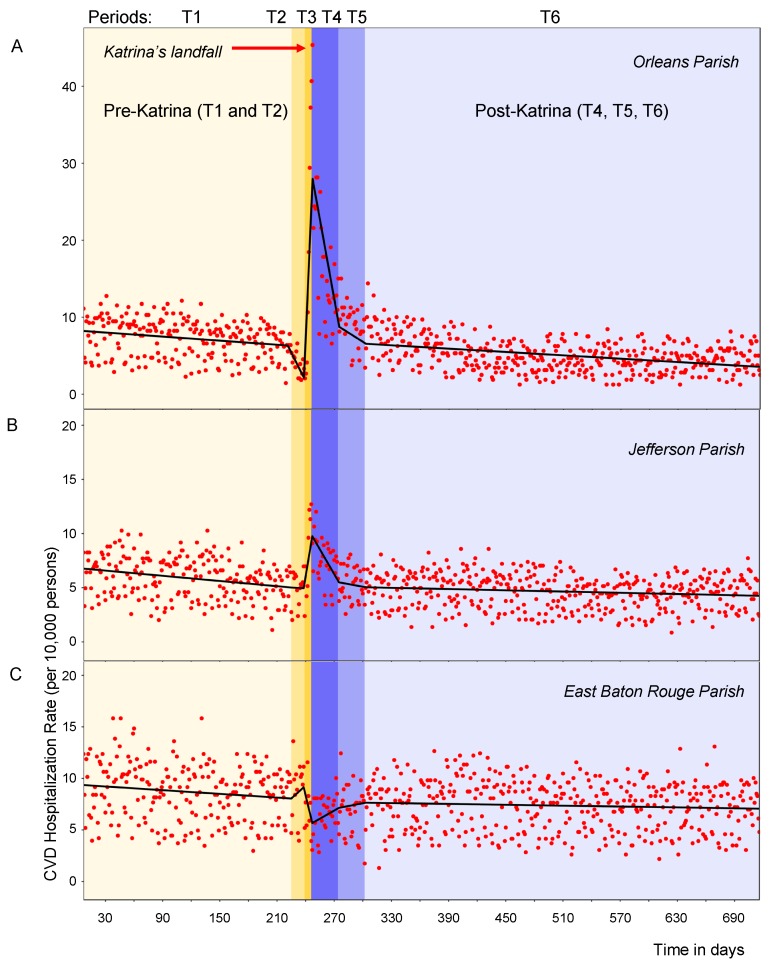
Time series of daily CVD hospitalization rates per 10,000 elderly adults with superimposed predicted values for 7 January 2005–17 December 2006 in Orleans Parish (**Panel A**), Jefferson Parish (**Panel B**), and East Baton Rouge Parish (**Panel C**). Actual daily hospitalization rates are displayed as points; rates predicted by a piecewise regression model are displayed as lines. The different hues and capital letters indicate the six distinct time periods with landfall occurred on 29 August 2005 right after day 240, at the border between T2 and T3 periods.

**Figure 4 ijerph-16-00074-f004:**
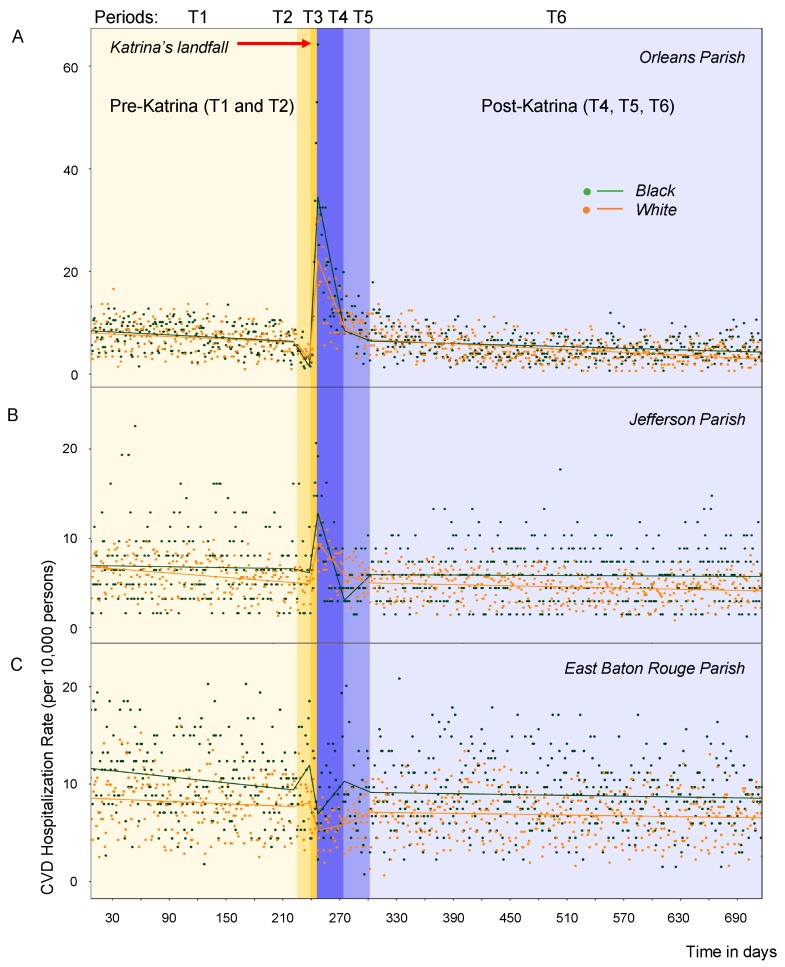
Time series of daily CVD hospitalization rates per 10,000 elderly adults with superimposed predicted values, grouped by race, for 7 January 2005–17 December 2006 in Orleans Parish (**Panel A**), Jefferson Parish (**Panel B**), and East Baton Rouge Parish (**Panel C**). Actual daily hospitalization rates are displayed as points; rates predicted using a piecewise regression model are displayed as lines: teal and orange lines indicate CVD rates for black and white older adults, respectively. The different hues and capital letters indicate the six distinct time periods with landfall occurred on 29 August 2005 right after day 240, at the border between T2 and T3 periods.

**Table 1 ijerph-16-00074-t001:** Six time periods (T1–T6) identified for in-depth evaluation.

Period	Ending Point Knot	Duration (Days)	Description
T1	Day 225 (13 August 2005)	219	The pre-Katrina period, before the storm or major warnings and evacuations.
T2	Day 238 (26 August 2005)	13	The two weeks before Hurricane Katrina, when officials began issuing warnings, and evacuations began.
T3	Day 247 (4 September 2005)	9	The week immediately following Hurricane Katrina, starting the day before landfall, and ending at the peak.
T4	Day 275 (2 October 2005)	28	The first month following the immediate period.
T5	Day 303 (30 October 2005)	28	The second months after the immediate period.
T6	Day 710 (17 December 2006)	413	Post-Katrina period of stabilization and recovery.

**Table 2 ijerph-16-00074-t002:** Demographic characteristics and cardiovascular disease (CVD) rates for three parishes of Louisiana for total, black and white older adults before (Period T1: 7 January–9 August 2005) and after (Periods T3 and T4: 27 August–2 October 2005) Hurricane Katrina.

Characteristics		Population Estimates (%)		Population Density (people/mi^2^)		Average Daily Cases		Average Dailyrates ± st. Deviation	
Period		T1	T3–T4	T1	T3–T4	T1	T3–T4	T1	T3–T4
Orleans	Total	48,599	31,968	139	91.4	35.4	54.4	7.3 ± 2.44	17.0 ± 10.23
	Black	26,648 (54.8%)	15,105 (47.6%)	76.2	43.2	19.7	29.9	7.4 ± 2.70	19.8 ± 13.70
	White	20,494 (42.2%)	16,126 (50.4%)	58.6	46.1	14.5	23.5	7.1 ± 2.82	14.5 ± 7.79 *
Jefferson	Total	54,513	58,256	111.7	119.4	32.3	43.5	5.9 ± 1.90	7.5 ± 2.58
	Black	6206 (11.3%)	6777 (11.6%)	12.7	13.9	4	5.4	6.5 ± 3.94	8.0 ± 5.05
	White	45,841 (84.1%)	49,359 (84.7%)	93.9	101.1	27.1	36.9	5.9 ± 1.90 **	7.5 ± 2.48
East Baton Rouge	Total	40,425	45,837	85.9	97.4	35.3	29.7	8.7 ± 2.74	6.5 ± 1.91
	Black	11,337 (28.0%)	13,436 (29.3%)	24.1	28.6	11.9	11.7	10.5 ± 4.00	8.7 ± 3.68
	White	28,174 (69.7%)	31,398 (68.5%)	59.9	66.7	22.8	17.7	8.1 ± 2.77 ***	5.6 ± 1.91

* *p* < 0.05, ** *p* < 0.01, *** *p* < 0.0001 for comparing Black and White population groups using *t*-test.

**Table 3 ijerph-16-00074-t003:** Summary statistics for CVD hospitalization rates (daily cases per 10,000 older adults) before (Periods T1 and T2), during (Period T3), and after Hurricane Katrina (Periods T4, T5, and T6) in three parishes for total, black and white older adults. Slopes and intercepts were generated using segmented regression models.

*Periods*		T1	T2	T3	T4	T5	T6
Number of days		219	13	9	28	28	413
Date Range		7 January 2005–13 August	14–26 August	27 August–4 September	5 September–2 October	3–30 October	31 October–17 December 2006
**Total population**							
**Orleans Parish**	Min, Max	1.44, 12.76	1.85, 6.58	2.06, 45.36	6.5, 28.15	3.4, 15.02	1.25, 14.39
	Mean ± SD	7.25 ± 2.44	3.91 ± 1.45 ^a^	18.46 ± 17.3 ^a^	13.76 ± 6.51 ^a^	9.54 ± 2.78	4.69 ± 2.08 ^a^
	Intercept ^b^	8.31	5.07	4.99	27.28	8.66	6.51
	Slope ± SE	−0.01 ± 0.003	−0.24 ± 0.06 ^c^	3.12 ± 0.19 ^c^	−3.56 ± 0.17 ^c^	0.61 ± 0.06 ^c^	0.07+0.03 ^c^
**Jefferson Parish**	Min, Max	1.10, 10.27	2.02, 6.97	2.38, 12.70	3.43, 12.02	2.57, 8.07	0.86, 8.58
	Mean ± SD	5.90 ± 1.90	5.01 ± 1.52	8.18 ± 3.70 ^a^	7.26 ± 2.15 ^a^	5.36 ± 1.53	4.65 ± 1.57 ^a^
	Intercept	6.82	5.02	5.49	9.60	5.48	5.04
	Slope ± SE	−0.01 ± 0.002	0.003 ± 0.05	0.54 ± 0.14 ^c^	−0.68 ± 0.11 ^c^	0.14 ± 0.04 ^c^	0.01 ± 0.02
**East BR Parish**	Min, Max	2.97, 15.83	3.96, 13.61	3.05, 11.56	2.84, 10.04	1.75, 12.44	1.31, 13.09
	Mean ± SD	8.69 ± 2.74	9.11 ± 2.69	6.52 ± 2.58	6.55 ± 1.70 ^a^	6.96 ± 2.42 ^a^	7.39 ± 2.37 ^a^
	Intercept	9.40	6.43	5.19	4.58	6.76	7.20
	Slope ± SE	−0.01 ± 0.003	0.09 ± 0.08	−0.47 ± 0.21 ^c^	0.44 ± 0.17 ^c^	−0.03 ± 0.06	−0.02 ± 0.02
**Black population**							
**Orleans Parish**	Min, Max	1.50, 13.51	1.13, 8.26	1.50, 64.22	7.94, 32.44	1.99, 15.23	0.66, 17.87
	Mean ± SD	7.37 ± 2.70	3.91 ± 2.12 ^a^	26.29 ± 23.7 ^a^	17.54 ± 8.00 ^a^	9.10 ± 3.51	5.30 ± 2.43 ^a^
	Intercept	8.50	4.94	5.16	33.60	8.37	6.40
	Slope ± SE	−0.01 ± 0.004	−0.28 ± 0.08 ^c^	3.96 ± 0.25 ^c^	−4.60 ± 0.22 ^c^	0.86 ± 0.08 ^c^	0.07 ± 0.03 ^c^
**Jefferson Parish**	Min, Max	1.61, 22.56	1.61, 16.11	1.61, 20.66	2.95, 16.23	1.48, 10.33	1.48, 17.71
	Mean ± SD	6.76 ± 3.94	6.58 ± 4.32	11.79 ± 6.81 ^a^	7.05 ± 3.78	4.86 ± 2.31	5.81 ± 2.95 ^a^
	Intercept	6.96	6.43	6.87	12.46	3.17	5.92
	Slope ± SE	−0.002 ± 0.004	−0.03 ± 0.08	0.77 ± 0.25 ^c^	−1.09 ± 0.22 ^c^	0.45 ± 0.08 ^c^	−0.10 ± 0.03 ^c^
**East BR Parish**	Min, Max	1.76, 20.29	5.29, 18.52	2.23, 13.40	4.47, 19.35	0.74, 20.10	1.49, 20.84
	Mean ± SD	10.48 ± 4.00	11.81 ± 4.65	7.30 ± 3.72	9.22 ± 3.60	8.45 ± 4.28	8.90 ± 3.55 ^a^
	Intercept	11.67	10.14	11.36	7.04	10.24	9.13
	Slope ± SE	−0.01 ± 0.004	0.16 ± 0.09	−0.70 ± 0.27 ^c^	0.68 ± 0.25 ^c^	−0.16 ± 0.09	0.04 ± 0.04
**White population**							
**Orleans Parish**	Min, Max	0.98, 16.59	1.95, 5.86	2.44, 30.39	4.96, 23.30	3.10, 15.50	0.62, 13.64
	Mean ± SD	7.09 ± 2.83	4.20 ± 1.05 ^a^	16.56 ± 11.7 ^a^	13.80 ± 6.15 ^a^	8.81 ± 2.89	4.76 ± 2.43 ^a^
	Intercept	8.06	5.29	5.17	21.75	8.96	6.66
	Slope ± SE	−0.01 ± 0.003	−0.18 ± 0.06 ^c^	2.32 ± 0.20 ^c^	−2.60 ± 0.18 ^c^	0.39 ± 0.06 ^c^	0.08 ± 0.03 ^c^
**Jefferson Parish**	Min, Max	1.31, 9.82	1.96, 6.76	2.62, 12.56	3.24, 11.95	2.23, 8.51	0.81, 8.71
	Mean ± SD	5.90 ± 1.90	5.00 ± 1.49	7.76 ± 3.34 ^a^	7.43 ± 2.21 ^a^	5.47 ± 1.72	4.55 ± 1.58 ^a^
	Intercept	6.84	4.99	5.40	9.34	5.86	4.99
	Slope ± SE	−0.01 ± 0.002	−0.0001 ± 0.04	0.52 ± 0.12 ^c^	−0.64 ± 0.11 ^c^	0.10 ± 0.04 ^c^	0.03 ± 0.02
**East BR Parish**	Min, Max	2.48, 15.97	3.19, 13.13	2.55, 10.51	1.91, 8.60	1.27, 10.19	0.64, 13.38
	Mean ± SD	8.09 ± 2.77	8.16 ± 2.73	6.17 ± 2.63	5.53 ± 1.65 ^a^	6.34 ± 2.26 ^a^	6.83 ± 2.40 ^a^
	Intercept	8.57	7.79	7.74	5.25	5.90	7.10
	Slope ± SE	−0.004 ± 0.003	0.03 ± 0.06	−0.33 ± 0.18	0.34 ± 0.16 ^c^	0.02 ± 0.06	−0.05 ± 0.02

*^a^* indicate *p*-value < 0.05 from *t*-tests comparing the average rate of each period to Period T1. *^b^* reflect the intercept or predicted value at beginning of the interval from piecewise regression models. *^c^* indicate *p*-value < 0.05 from *t*-tests for slope significance.

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
