# Peer review of "Cardiovascular Disease Hospitalizations in Louisiana Parishes’ Elderly before, during and after Hurricane Katrina"

_ijerph, 2018, doi:10.3390/ijerph16010074_

Round 1
Reviewer 1 Report
This is a temporal and spatial analysis of CVD hospitalization using CMMS data focusing on the impact of Hurricane Katrina in the three representative parishes in Lousiana. The results are clearly indicating age, race, socioeconomic status are the social determinants of health impacts of the disaster. It might be very difficult to perform multivariate analysis using these factors, but the robustness of the large size public health database gives an insight to our understanding of the health risks in disaster. Clarification of the following points are necessary.
1. Line 136: "CVD" not "CDV"
2. Line 146: Where is Table 2S?
3. Line 198: Where is Table S3?
4. Line 203: Are supplemental Tables named S#, or #S?
5. Figure 1: Please indicate the time frame in the legend or figure as stated in Line 239: Panels C and D (T1), Panels E and F (T3 and T4)
6. Line 247: Indicate the target of comparison, i.e. vs. T1, vs. Black population, or vs. to other parish?
7. Line 275: Is "0.003 cases per day" correct? Not 3 cases/day, or do you mean the slope between T2 and T3 (0.54 with c)?
8. Line 296: Indicate the target of control. ex. "p<0.001 vs. T1)
9. Line 297: Indicate that these numbers are from Table 2 and add information of target of comparison in the footnotes of Table 2.
10. Line 315: There is no definition of Period B even though this is the description in Ref. 3. I think two days before Katrina is sufficient.
11. Line 367: Do you think this is the reason why in all Parishes in total, black and white the rate of CVD significantly decreased in T6 as indicated in Table 3? If not, please discuss why the decline in T6 occurred. It might partly be the situation described in Lines 411-423, but make a clearer discussion on ths particular point because it might include some hint for "Building Back Better" and the unanswered questions.
12. Line 447: English might lack subject.
13. Line 476: Table "1S", or "S1"? Within the manuscript, Supplementary Tables 2S (line 146), S3 (Line 198) are lacking.
Author Response
''''''''''''''''''''''''''''''''''''''''''''''''''''''''''''''''''''''''''''''''''''''''''''''''''''''''''''''''''''''''''''''''''''''''''''''''''''''''''''''''''''''''''''''''''''''''''''''''''''''''''''''''''''''''''''''''''''''''''''''''''''''''''''''''''''''''''''''''''''''''''''''''''''''''''''''''''''''''''''''''''?
1. Line 136: "CVD" not "CDV"
- corrected
2. Line 146: Where is Table 2S?
- We apologize for omission, the Table S2 has been added
3. Line 198: Where is Table S3?
- We apologize for omission, the Table S3 has been added
4. Line 203: Are supplemental Tables named S#, or #S?
- corrected
5. Figure 1: Please indicate the time frame in the legend or figure as stated in Line 239: Panels C and D (T1), Panels E and F (T3 and T4)
– corrected
6. Line 247: Indicate the target of comparison, i.e. vs. T1, vs. Black population, or vs. to other parish?
– corrected
7. Line 275: Is "0.003 cases per day" correct? Not 3 cases/day, or do you mean the slope between T2 and T3 (0.54 with c)?
– corrected
8. Line 296: Indicate the target of control. ex. "p<0.001 vs. T1)
– corrected
9. Line 297: Indicate that these numbers are from Table 2 and add information of target of comparison in the footnotes of Table 2.
– corrected
10. Line 315: There is no definition of Period B even though this is the description in Ref. 3. I think two days before Katrina is sufficient.
– corrected
11. Line 367: Do you think this is the reason why in all Parishes in total, black and white the rate of CVD significantly decreased in T6 as indicated in Table 3? If not, please discuss why the decline in T6 occurred. It might partly be the situation described in Lines 411-423, but make a clearer discussion on ths particular point because it might include some hint for "Building Back Better" and the unanswered questions.
- We clarified that we don’t know the reasons for observed steady declines in CVD rates post-Katrina, it may include general population displacement especially the older adults.
12. Line 447: English might lack subject.
– corrected
13. Line 476: Table "1S", or "S1"? Within the manuscript, Supplementary Tables 2S (line 146), S3 (Line 198) are lacking.
– We apologize for omission, the Tables S2 and S3 have been added
Reviewer 2 Report
Review report for International J. of Environmental Research and Public Health Nov 20 2018 The paper is well written and addresses an important problem. Main comments Since the paper focuses on elders, it should be in the title. The abstract says “The research on how health and health care disparities impact response to and recovery from a disaster.” However, the paper is based on hospitalization data and I don’t see its connection with “response to and recovery”. Some clarifications are needed. In abstract, “We compared CVD hospitalization ...” does not mention elderly. I think it should be added. The abstract says “In the two parishes … 7.25...18.5...”. The numbers are also for Orleans parish. It says “Disparities in CVD… in all three parishes”. I don’t see the disparity for Orleans Parish prior to Hurricane Katrina. P3. More details should be provided for the datasets including proper references. P3, ln. 138: For all numbers of CVD hospitalization records, are they for people aged 65+ (e.g., 17,769 for Orleans Parish) ? There are several places where the authors might be referring to people aged 65+ but does not explicit mention them. Readers could be confused. P4, Ln. 149-1More descriptions of American Community Survey and intercensual estimate are needed. What is the resolution of the data? By days? It does not seem feasible. How did the author get estimate of population for a segment of a year? P4, ln. 169: A reference for “loess smoother” is needed. P5, ln.198: Table S3 is unavailable. P5, ln. 212: add East Baton Rouge. What is the time reference for Fig. 1? Based on the text, it seems that Panel A is based on T1, T3, and T4, Panel C is based on T3. Some clarifications are needed. Average daily elderly CVD rates per county are shown in Fig. 1. But there is little discussion in the text. P7, ln. 240: “...17 cases per 10,000 people aged 65+... Table 2)”. Where in Table 2 is it shown? P11, ln. 297: “...19.8...14.5…”. Are these numbers in Table 3? Maybe highlight them. P11, ln. 308: “White adults also…”. Is this discussion about Orleans Parish? P.12, ln.317: should “New Orleans” be “Orleans”? Ln. 324: “Figure 1” does not show what the text described (i.e., black residents). Are there any statistical tests for the difference in increase in CVD hospitalization between the black and white? Minor There are some grammar errors. For example, P7, ln. 242: “prior” ->”Prior to”. P12, ln. 340: define acronyms at first use; e.g., PTSD.
Author Response
1. Since the paper focuses on elders, it should be in the title.
– We modified the title.
2. The abstract says “The research on how health and health care disparities impact response to and recovery from a disaster.” However, the paper is based on hospitalization data and I don’t see its connection with “response to and recovery”. Some clarifications are needed.
– We believe that hospitalization rates months after Katrina reflects the community recovery in this regard.
3. In abstract, “We compared CVD hospitalization ...” does not mention elderly. I think it should be added.
– In the abstract, the fifth sentence “We abstracted 383,552 CVD hospitalization records for Louisiana’s patients aged 65+ in 2005–6 from the database maintained by the Center of Medicare & Medicaid Services.” explicitly stated the age of patients.
4. The abstract says “In the two parishes … 7.25...18.5...”. The numbers are also for Orleans parish. It says “Disparities in CVD… in all three parishes”. I don’t see the disparity for Orleans Parish prior to Hurricane Katrina.
- We modified the sentence.
5. P3. More details should be provided for the datasets including proper references.
- We offered 7 references of the use of the CMS data set and added one more.
6. P3, ln. 138: For all numbers of CVD hospitalization records, are they for people aged 65+ (e.g., 17,769 for Orleans Parish)? There are several places where the authors might be referring to people aged 65+ but does not explicit mention them. Readers could be confused.
- We clarified the text.
7. P4, Ln. 149-1More descriptions of American Community Survey and intercensual estimate are needed. What is the resolution of the data? By days? It does not seem feasible. How did the author get estimate of population for a segment of a year?
– We provided additional description in the text on how we had linearly interpolated information on population by county or parish to estimate daily rates for two segments before and after Katrina based on available estimates for 2005 and 2006.
8. P4, ln. 169: A reference for “loess smoother” is needed.
- Add a sent about loess with refs, check [33,34?]
9. P5, ln.198: Table S3 is unavailable.
– We apologize for omission, the Table S3 have been added.
10. P5, ln. 212: add East Baton Rouge.
- Corrected.
11. What is the time reference for Fig. 1? Based on the text, it seems that Panel A is based on T1, T3, and T4, Panel C is based on T3. Some clarifications are needed. Average daily elderly CVD rates per county are shown in Fig. 1. But there is little discussion in the text.
– We added time periods to the legend of Figure 1.
12. P7, ln. 240: “...17 cases per 10,000 people aged 65+... Table 2)”. Where in Table 2 is it shown?
- We clarified in the text “As shown in Table 2, Orleans Parish had a notably high average daily CVD hospitalization rate of 17.010.23 cases/day per 10,000 people aged 65+ after the storm (Periods T3 and T4), while the number of residents declined by almost 7,000 people.”
13. P11, ln. 297: “...19.8...14.5…”. Are these numbers in Table 3? Maybe highlight them.
- We clarified in the text that these numbers were shown in Table 2 and highlighted the numbers.
14. P11, ln. 308: “White adults also…”. Is this discussion about Orleans Parish?
- We clarified the text by adding “In the most affected Orleans Parish,”
15. P.12, ln.317: should “New Orleans” be “Orleans”?
- Corrected.
16. Ln. 324: “Figure 1” does not show what the text described (i.e., black residents).
- Reference to Figure 1 has been removed.
17. Are there any statistical tests for the difference in increase in CVD hospitalization between the black and white?
- We clarified that in Table 2 T-test was used for comparing Black and White population groups.
Minor
There are some grammar errors. For example,
18. P7, ln. 242: “prior” ->”Prior to”.
- Corrected
19. P12, ln. 340: define acronyms at first use; e.g., PTSD.
Corrected
Reviewer 3 Report
Overall, I found the paper to be very interesting and a significant contribution to the literature. My perspective in reading this paper comes from my expertise in public health aspects of disasters with a focus in gulf coast disasters. I am not familiar with Loess smoother analysis. So my comments here focus on the disaster and public health perspective as well as GIS.
A strength of the manuscript is the use of CMS data covering a wide range of time from before, during, and after Hurricane Katrina. A lack of pre-event data is often a limitation in disaster research and using CMS data has enabled you to address this limitation effectively. The discussion of findings makes a number of important recommendations for improving public health practice in a post-disaster setting.
A few recommendation to improve the manuscript:
More details on the GIS mapping would be helpful for the reader. What geography did you map the ACS data to in GIS? And how was the community level data accounted for in the modeling?
While much of the article correctly discusses Louisiana's Parishes, there are a number of places, especially in the figures/tables where County is mistakenly used.
A careful proofreading for tense/plurals is needed throughout. In my read, I circled nine typos of this nature
Finally, one additional suggested article to read that relates looks at CVD among other chronic diseases and Hurricane Katrina:
Sharma, A. J., Weiss, E. C., Young, S. L., Stephens, K., Ratard, R., Straif-Bourgeois, S., ... & Rubin, C. H. (2008). Chronic disease and related conditions at emergency treatment facilities in the New Orleans area after Hurricane Katrina. Disaster medicine and public health preparedness, 2(1), 27-32.
Author Response
1. More details on the GIS mapping would be helpful for the reader. What
geography did you map the ACS data to in GIS?
- We used county (or parish) level data, as shown in Supplemental Table S4.
And how was the community level data accounted for in the modeling?
- We clarified in the text: For spatial mapping county-specific hospitalization rates were calculated by dividing counts of CVD cases per day for all period, pre-Katrina and post-Katrina periods by the parish population (65+) with the multiplier of 100,000. For temporal analysis, for each of the three parishes the daily hospitalization rates were calculated by dividing counts of CVD cases per day by the interpolated estimates of parish population with the multiplier of 10,000.”
2. While much of the article correctly discusses Louisiana's Parishes, there are a
number of places, especially in the figures/tables where County is mistakenly
used.
- We clarified in the Methods section “that in Louisiana, a “parish” is equivalent to a “county,” which is a more generic term, so both terms can be used interchangeably.”
3. A careful proofreading for tense/plurals is needed throughout. In my read, I
circled nine typos of this nature.
- We corrected as many as we can find.
4. Finally, one additional suggested article to read that relates looks at CVD
among other chronic diseases and Hurricane Katrina:
Sharma, A. J., Weiss, E. C., Young, S. L., Stephens, K., Ratard, R., Straif-
Bourgeois, S., ... & Rubin, C. H. (2008). Chronic disease and related
conditions at emergency treatment facilities in the New Orleans area after
Hurricane Katrina. Disaster medicine and public health preparedness, 2(1), 27-
- We are thankful for the suggested reference and included it in the text.
Round 2
Reviewer 2 Report
I appraise the authors for addressing all the comments. There are two more comments.
12. P7, ln. 240: “...17 cases per 10,000 people aged 65+... Table 2)”. Where in Table 2 is it shown?
-We clarified in the text “As shown in Table 2, Orleans Parish had a notably high average daily CVD hospitalization rate of 17.010.23 cases/day per 10,000 people aged 65+ after the storm (Periods T3 and T4), while the number of residents declined by almost 7,000 people.”
Did you mean “17,000” instead of “7,000’?
17. Are there any statistical tests for the difference in increase in CVD hospitalization between the black and white?
- We clarified that in Table 2 T-test was used for comparing Black and White population groups.
Why are “*” appear in sometimes in the column for Black people and other times in the column for white people in Table 2?
Author Response
Dear Reviewers,
We have corrected two typos as follow:
1. Did you mean “17,000” instead of “7,000’?
Yes, the number should be 17,000. Thank you for noting.
2. Why are “*” appear in sometimes in the column for Black people and other times in the column for white people in Table 2?
Again, thank you for noting. Now "*" has been moved to the correct position.
We have highlighted corrected text in red in the text attached and will resubmit shortly.
Best regards,
Elena N. Naumova, Ph.D.
Professor and Chair, Division of the Nutrition Data Science
Friedman School of Nutrition Science and Policy, Tufts University